# Heterogeneity of Phenotypic and Functional Changes to Porcine Monocyte-Derived Macrophages Triggered by Diverse Polarizing Factors In Vitro

**DOI:** 10.3390/ijms24054671

**Published:** 2023-02-28

**Authors:** Giulia Franzoni, Lorena Mura, Elisabetta Razzuoli, Chiara Grazia De Ciucis, Floriana Fruscione, Filippo Dell’Anno, Susanna Zinellu, Tania Carta, Antonio G. Anfossi, Silvia Dei Giudici, Simon P. Graham, Annalisa Oggiano

**Affiliations:** 1Department of Animal Health, Istituto Zooprofilattico Sperimentale della Sardegna, 07100 Sassari, Italy; 2Department of Biomedical Sciences, School of Medicine, University of Sassari, 07100 Sassari, Italy; 3National Reference Center of Veterinary and Comparative Oncology (CEROVEC), Istituto Zooprofilattico Sperimentale del Piemonte, Liguria e Valle d’Aosta, 16129 Genova, Italy; 4Department of Veterinary Medicine, University of Sassari, 07100 Sassari, Italy; 5The Pirbright Institute, Ash Road, Pirbright, Woking GU24 ONF, UK

**Keywords:** pig, macrophages, polarization, classical activation, IL-4, IL-10, TGF-β, dexamethasone, cytokines, surface markers, Toll-like receptors

## Abstract

Swine are attracting increasing attention as a biomedical model, due to many immunological similarities with humans. However, porcine macrophage polarization has not been extensively analyzed. Therefore, we investigated porcine monocyte-derived macrophages (moMΦ) triggered by either IFN-γ + LPS (classical activation) or by diverse “M2-related” polarizing factors: IL-4, IL-10, TGF-β, and dexamethasone. IFN-γ and LPS polarized moMΦ toward a proinflammatory phenotype, although a significant IL-1Ra response was observed. Exposure to IL-4, IL-10, TGF-β, and dexamethasone gave rise to four distinct phenotypes, all antithetic to IFN-γ and LPS. Some peculiarities were observed: IL-4 and IL-10 both enhanced expression of IL-18, and none of the “M2-related” stimuli induced IL-10 expression. Exposures to TGF-β and dexamethasone were characterized by enhanced levels of TGF-β2, whereas stimulation with dexamethasone, but not TGF-β2, triggered CD163 upregulation and induction of CCL23. Macrophages stimulated with IL-10, TGF-β, or dexamethasone presented decreased abilities to release proinflammatory cytokines in response to TLR2 or TLR3 ligands: IL-10 showed a powerful inhibitory activity for CXCL8 and TNF release, whereas TGF-β provided a strong inhibitory signal for IL-6 production. While our results emphasized porcine macrophage plasticity broadly comparable to human and murine macrophages, they also highlighted some peculiarities in this species.

## 1. Introduction

Macrophages are innate immune cells which were discovered in the late nineteenth century by Metchnikoff and named due to their phagocytic activity (“macro” (big) “phage” (eaters)) [1]. Later, it was observed that macrophages are involved in a wide array of functions, in tissue homeostasis, by clearing senescent cells, cellular debris, and repairing tissues after inflammation [2] and also in immune responses to infective and not infective stressors [3].

Macrophages are characterized by remarkable plasticity, and they can quickly change their function and phenotype in response to external stimuli [4]. The two antithetic extremes of activation states are represented by classically activated (M1) macrophages, characterized by increased microbicidal or tumoricidal capacity, and alternatively activated (M2) macrophages, associated with mechanisms of immunosuppression and wound repair [5]. In humans and mice, M2 macrophages have been generated in vitro by exposure to IL-4 and/or IL-13, whereas classical macrophage activation has been achieved in vitro by exposure to two signals: the first signal is the obligatory cytokine IFN-γ, whereas the second signal is TNF (itself or an TNF inducer). TLR agonists such as LPS can induce endogenous TNF production, and therefore, they are frequently used as the second signal to achieve classical activation [5]. 

This simplistic view of two potential statuses was subsequently refined with alternative activated macrophages being divided into subsets, such as M2a macrophages (following stimulation with IL-4 or IL-13), M2b macrophages (following exposure to immune complexes in combination with IL-1β or LPS), and M2c macrophages (stimulated with IL-10, TGF-β, or a glucocorticoid) [6]. Considering that exposure to diverse activators can lead to unique phenotypes [4,7], nomenclature based on the activator/s used, for example, M(IL-4), M(IFN-γ), M(IL-10), M(LPS), and M(Ig), has also been proposed [8].

Pigs share some anatomical and physiological similarities with humans, especially in the digestive, urinary, integumentary, and immune systems [9,10]. These similarities, combined with their manageable behavior and size, mean that pigs have been widely used in translational studies, such as preclinical evaluation of vaccine candidates and therapeutics [11,12] and preclinical toxicologic testing of pharmaceuticals or other chemicals [9,10]. The porcine model has been particularly relevant in studies focused on human sexually transmitted infection [13], as well as in nanomedicine-based studies [14,15].

Several studies have suggested that pig models are better than mouse models for understanding human innate immunity, and pigs have presented higher predictive values than rodents in preclinical studies [16]. For example, it has been described that porcine macrophages resemble human macrophages in their response to LPS, with a similar inducible gene expression profile [17,18]. Macrophage polarization in pigs has not been extensively analyzed. A better understanding of porcine macrophage polarization could help to improve translational studies and could aid the interpretation of in vitro and in vivo studies of host–pathogen interactions. In order to better benefit translational studies using this large animal model, we performed a detailed characterization of porcine macrophages following exposure to different polarizing stimuli.

## 2. Results

The ability of IFN-γ and LPS (classical activation) and “M2-related” polarizing factors to modulate porcine moMΦ phenotype and functionality was assessed though an integrative analytical approach, spanning microscopy, flow cytometry, multiplex ELISA, RT-qPCR, and qPCR array. 

### 2.1. Impact of Diverse Polarizing Factors on Porcine moMΦ Phenotype

Monocyte-derived macrophages (moMΦ) were left untreated, or stimulated with IFN-γ + LPS to generate classically activated macrophages (moM1). In parallel, moMΦ were stimulated with “M2 polarizing factors”, i.e., IL-4, IL-10, TGF-β, or dexamethasone. Twenty-four hours post-stimulation, the phenotypes of macrophage subsets were investigated with microscopy and flow cytometry. 

Microscopy revealed that all macrophage subsets presented with a spherical shape with short “hairy” protrusions on their surface (Figure 1 and Appendix A), in agreement with our previous work [19,20]. We observed that 24 h treatment with IFN-γ and LPS, IL-4, IL-10, TGF-β, or dexamethasone did not alter the dimension or granularity of the moMΦ (Figure 1), in agreement with our previous work [19,20].

Flow cytometry was employed to determine the phenotypic differences between macrophage subsets. Classical activation (IFN-γ and LPS) resulted in upregulation of MHC I, MHC II DR, and CD169, the last two both in terms of percentages of positive cells and mean fluorescence intensity (MFI) of positive cells (Figure 2 and Appendix A). IL-4 did not modulate expression of the tested surface markers, except for a downregulation of CD14 (in terms of MFI of positive cells), and CD163 (decrease percentage of positive cells), although the latter without statistical significance (Figure 2 and Appendix A). Stimulation with IL-10 modulated the surface expression of MHC I, MHC II DR, CD14, CD16, and CD163. As displayed in Figure 2 and Appendix A, IL-10 induced downregulation of CD14, and upregulation of CD163 and CD16 (all in terms of MFI), in agreement with our previous work [20]. Interestingly, in this study, we observed that IL-10 significantly upregulated MHC I but downregulated MHC II DR expression (Figure 2). In agreement with our previous work [20], we observed that TGF-β downregulated expression of CD14, MHC II DR, and CD163, but did not alter expression of MHC I and CD169 (Figure 2 and Appendix A). Stimulation of moMΦ with dexamethasone resulted in MHC II DR downregulation (MFI of positive cells), but enhanced expression of CD163 and CD14, both in terms of percentages of positive and MFI of positive cells (Figure 2 and Appendix A). 

### 2.2. Induction of Cytokine Expression and Release by moMΦ in Response to Diverse Polarizing Factors

To evaluate how macrophage stimulation with IFN-γ and LPS or diverse M2-related polarizing factors (IL-4, IL-10, TGF-β, and dexamethasone) modulated innate immunity, the RT2 Profiler PCR Array System covering 84 porcine cytokine and chemokine genes was employed: expression of several proinflammatory and anti-inflammatory interleukins (IL), chemokines, interferons (IFN), and members of the tumor necrosis factor family genes were investigated. The gene expression in each group was first normalized to the untreated control (moMΦ), and in Figure 3 up- and downregulated cytokine genes are presented. For each gene, the fold change normalized to the untreated control and the corresponding *p*-value are presented (Figure 3 and Appendix A); fold changes >2.0 and *p*-value < 0.05 were considered to be significant variations. Scatter plots presenting fold changes of all 84 genes in each macrophage subsets compared to the untreated control are presented in Appendix A, whereas the unsupervised hierarchical clustering analysis of gene expression changes in moMΦ stimulated with diverse polarizing factors is presented in Appendix A.

Our results showed that stimulation with IFN-γ and LPS resulted in upregulation with fold change >2.0 of cytokine genes, including AMCF-II, CCL17, CCL19, CCL2, CCL20, CCL22, CCL3L1, CCL4, CCL5, CCL8, CSF2, CSF3, CXCL10, CXCL11, LOC396594, CXCL9, FASLG, IFNB1, IFNG, IL10, IL12A, IL12B, IL15, IL17F, IL18, IL1α, IL1β, IL2, IL22, IL23A, IL27, IL4, IL6, IL7, CXCL8, INHBA, LIF, LOC100515857, CCL23, CCL16, CXCL13, LTA, LTB, MSTN, SPP1, TNF, TGFβ1, TNFSF10, and VEGFA, with statistical significance (*p* < 0.05) for CCL2, CCL5, CCL8, CXCL10, LOC396594, IL10, IL7, and SPP1. Only three genes were downregulated in moM1 compared to the untreated control with fold change >2.0: CCL21 (*p* = 0.053818), IFN-ALPHA5 (*p* = 0.659246), and TGFβ2 (*p* = 0.030825) (Figure 3, Appendix A).

Instead, stimulation with IL-4 resulted in enhanced expression (fold change > 2.0) of ADIPQ, BMP2, CCL17, CCL22, CCL3L1, CCL8, FASLG, IL18, IL27, IL6, CCL16, CXCL13, and TNFSF10, although with statistical significance only for BMP2 (*p* = 0.041345), and IL18 (*p* = 0.0362872). Six genes were downregulated in moM(IL-4) compared to the untreated control (fold changes > 2.0): CCL20, CCL21, CSF1, IL1β, IL2, and SSP1, all without statistical significance. A *p*-value < 0.1 was observed for CCL21 (*p* = 0.065265) (Figure 3, Appendix A). 

Our data revealed that stimulation with IL-10 led to enhanced expression (fold changes > 2.0) of CCL8, LOC396594, IL18, CXCL8, CXCL13, and TNFSF13B, with statistical significance only for LOC396594 (*p* = 0.018069), IL18 (*p* = 0.013221), and TNFSF13B (0.002243). Several cytokine genes were instead downregulated (with fold change > 2.0) in moM(IL-10) compared to the untreated control: ADIPOQ, CCL17, CCL2, CCL20, CCL21, CCL22, CCL3IL1, CCL4, CXCL10, CXCL11, CXCL9, IFN-ALPHA-5, IFNβ1, IL12β, IL13, IL1α, IL1β, CCL24, and MSTN, although none with statistical significance (Figure 3, Appendix A). 

Stimulation with TGF-β triggered significantly enhanced expression of BMP2, BMP3, CCL21, and TGF-β2, although with statistical significance only for TGF-β2 (*p* = 0.022099). Several cytokine genes were downregulated in moM(TGF-β) compared to the untreated control (fold change > 2.0): ADIPQ, AMCF-II, CCL1, CCL17, CCL20, CCL22, CCL3L1, CCL4, CCL8, CSF2, CXCL10, CXCL11, CXCL9, IFN-ALPHA-5, IFNβ1, IL13, IL15, IL18, IL1α, IL2, IL27, CXCL13, CCL24, LTA, and TNF, although with statistical significance only for IL15 (*p* = 0.0481119) (Figure 3, Appendix A). 

Stimulation with dexamethasone resulted in substantial upregulation of only two cytokine genes (fold change > 2.0): CCL23 (*p* = 0.008826) and CCL16 (*p* = 0.064898), whereas it triggered downregulation (fold change > 2.0) of 32 cytokine genes: ADIPQ, CCL1, CCL17, CCL2, CCL20, CCL22, CCL3L1, CCL4, CCL8, CD40LG, CSF1, CSF2, CXCL10, CXCL11, CXCL9, FASLG, IL18, IL1α, IL1β, IL2, IL4, IL7, CXCL8, INHBA, LIF, CXCL13, LTA, LTB, SPP1, TNF, and TNFSF10, although without statistical significance; a *p*-value < 0.1 was observed for IL18 (*p* = 0.063833), IL1α (*p* = 0.081704), and SPP1 (*p* = 0.084537) (Figure 3, Appendix A). 

Quantitative RT-PCR was then employed to investigate gene expression of selected cytokines over time (4, 8, and 24 h post-stimulation). First, we monitored induction of two major anti-inflammatory cytokines: IL-10 and IL-1Ra. In our previous study [20], we surprisingly observed that IL-10 expression was not released or induced in response to stimulation with IL-4, IL-10, or TGF-β. In this study, we observed that IL-10 was enhanced in moM1 compared to the untreated control, but was not enhanced following stimulation with the other cytokines (Figure 3 and Figure 4). In addition, RT-PCR data revealed that IL-4, IL-10, TGF-β, and dexamethasone induced IL-10 downregulation (Figure 4). We observed that stimulation with IFN-γ and LPS also induced expression of IL-1Ra at all tested timepoints, whereas it was upregulated at 4 h post IL-4 response (Figure 4). TGF-β stimulation resulted in a small but statistically enhanced expression of IL-1Ra, whereas it was downregulated by stimulation with IL-10 and dexamethasone (Figure 4). 

Modulation of two other members of the IL-1 superfamily, IL-1β and IL-18, was monitored. As expected, stimulation with IFN-γ and LPS enhanced expression of the proinflammatory IL-1β at all tested timepoints, whereas downregulation was observed in the other subsets. A similar trend was observed for two other proinflammatory cytokines, IL-6 and TNF (Appendix A). In agreement with the array data, RT-PCR results showed that IL-18 was upregulated in moM(IL4) and moM(IL10) compared to the untreated control (moMΦ), whereas the expression of this IL-18 was downregulated after stimulation with TGF-β (24 h) or dexamethasone (all timepoints) (Figure 4). 

Finally, we monitored expression of the chemokines CXCL13 and CCL23, and the TGF-β superfamily member TGF-β2. Array data revealed that CXCL13 expression was enhanced following 24 h stimulation with either IFN-γ + LPS, IL4, or IL10, although without statistical significance. Nevertheless, a 388.82-fold change was observed in moM(IL-10) compared to the untreated control, with a *p*-value of 0.084008 (Appendix A). The expression of this chemokine was monitored over time on five different pigs and RT-PCR data showed that IFN-γ + LPS, IL-4, or IL-10 enhanced expression of this chemokine, at all tested timepoints (Figure 5). On the contrary, we observed CXCL13 downregulation 24 h post-stimulation with TGF-β and dexamethasone (Figure 5). Instead, we observed that CCL23 was significantly upregulated following stimulation with dexamethasone (Appendix A); thus, RT-PCR was employed to monitor expression of this chemokine over time. Although we observed that this chemokine was upregulated following dexamethasone stimulation, upregulation was also observed after stimulation with IFN-γ + LPS (Figure 5). TGF-β2 was significantly upregulated 24 h post-stimulation with TGF-β, whereas it was downregulated in response to classical activation (IFN-γ + LPS). In agreement, RT-PCR data showed that IFN-γ + LPS triggered TGF-β2 downregulation, whereas TGF-β and dexamethasone both enhanced its expression of all tested timepoints (Figure 5).

Multiplex ELISA was used to evaluate cytokine content in culture supernatants of moMΦ stimulated with diverse polarizing factors (24 h post-stimulation). In agreement with both array and qPCR results, we observed that stimulation with IFN-γ + LPS resulted in enhanced release of several proinflammatory cytokines: IL-1α, IL-1β, IL-6, CXCL8, IL-12, and TNF (Figure 6). A weak release of IL-18 was also detected in response to the IFN-γ + LPS treatment (Figure 6). Production of these cytokines was not observed in response to the either IL-4, IL-10, TGF-β, or dexamethasone treatment, with the exception of a small but statistically significant release of CXCL8 in response to IL-10 stimulation (Figure 6). In agreement with the gene expression data, small amounts of IL-10 were detected in culture supernatant of moMΦ stimulated with IFN-γ and LPS, but not following stimulation with either IL-4, TGF-β, or dexamethasone (Figure 6). In agreement with our previous work [20], significant higher levels of IL-10 were detected in the supernatants of IL-10-stimulated moMΦ; however, the amount detected 24 h post-stimulation (5.28 ± 1.68 ng/mL) was below the amount added to culture media at time 0 (20 ng/mL), suggesting that there was no de novo synthesis of this cytokine (Figure 6). A significant release of IL-1Ra was observed in response to stimulation with IFN-γ + LPS, but not IL-10, TGF-β, or dexamethasone (Figure 6). A small but statistically significant release of IL-1Ra was also observed in response to IL-4 stimulation (Figure 6).

### 2.3. Impact of Diverse Polarizing Factors on Porcine moMΦ Functionality 

We employed multiplex ELISA to evaluate the impact of diverse polarizing factors on subsequent macrophage responses to Toll-like receptor (TLR) agonists. The moMΦ cells were left untreated, or they were stimulated with IFN-γ + LPS, IL-4, IL-10, TGF-β, or dexamethasone. Then, 24 h later, supernatants were removed, and cells were left untreated or activated with TLR ligands. After 24 h, culture supernatants were collected, and levels of proinflammatory and anti-inflammatory cytokines were determined using multiplex ELISA. As expected, TLR2 and TLR3 genes were both highly expressed in porcine moMΦs (Appendix A), and thus ligands against both receptors were used in this study: the diacylated lipopeptide MagPam2Cys_P80 was used as a TLR2 ligand [21], and polyinosinic-polycytidylic acid (Poly I:C) was employed as a TLR3 ligand. Levels of proinflammatory (IL-1α, IL-1β, IL-6, CXCL8, IL-12, and TNF) or anti-inflammatory (IL-10 and IL-1Ra) in culture supernatants of macrophage subsets untreated or stimulated with MagPam2Cys_P80 or Poly I:C are presented in Figure 7 and Figure 8, respectively. 

We observed higher levels of IL-1α, Il-1β, CXCL8, and IL-12, in culture supernatants of macrophages stimulated with IFN-γ + LPS (moM1) compared to the untreated control (moMΦ) in the absence of subsequent stimulation. This was expected, because stimulation with IFN-γ + LPS triggered release of several proinflammatory cytokines, as presented in Figure 6, and release of some of them continued beyond 24 h post-stimulation. MoM1 presented an enhanced ability to release IL-12 compared to the untreated control (moMΦ) in response to either MagPam2Cys_P80 or Poly I:C (Figure 7 and Figure 8). Interestingly, our data revealed that moM1 possessed a reduced ability to release TNF in response to both of the agonists (Figure 7 and Figure 8). This might be linked to the reduced expression of TLR2 and TLR3 genes in moM1 compared to the untreated control (moMΦ) at the time of treatment with TLR agonists (24 h post-stimulation with IFN-γ + LPS) (Appendix A).

Stimulation with IL-4 did not statistically significantly impair the ability of moMΦs to release IL-1α, IL-1β, IL-6, CXCL8, and IL-12 in response to MagPam2Cys_P80 lipopeptide (Figure 7), although a trend was observed for IL-1β and IL-12. Nevertheless, moM(IL-4) presented a statistically significant lower ability to release TNF in response to the TLR2 agonist compared to the untreated control (moMΦ). This may be linked to the reduced expression of TLR2 in moM(IL-4) compared to the untreated control (moMΦ) at the time of treatment with MagPam2Cys_P80 lipopeptide (Appendix A). Treatment with this cytokine did not alter macrophage ability to release proinflammatory cytokines in response to Poly I:C (Figure 8). 

In accordance with our previous study [20], we observed that MoM(IL-10) presented a reduced ability to release IL-1α, IL-1β, IL-6, CXCL8, IL-12, and TNF in response to MagPam2Cys_P80 compared to the untreated control (moMΦ), although without statistical significance only for IL-1α, IL-1β, and IL-6 (Figure 7). Treatment with this immunosuppressive cytokine also resulted in a statistically significant reduced ability to release IL-12 and TNF in response to Poly I:C stimulation (Figure 8).

Stimulation with TGF-β did not statistically significantly impair the ability of moMΦs to release IL-1α and IL-1β, in response to MagPam2Cys_P80 lipopeptide (Figure 7), in agreement with our previous work [20]. Nevertheless, moM(TGF-β) presented a reduced ability to release IL-6, IL-12, and TNF in response to the TLR2 agonist compared to the untreated control (moMΦ), although without statistical significance for IL-12 (Figure 7). TGF-β treatment did not alter the release of CXCL8 in response to either MagPam2Cys_P80 or Poly I:C (Figure 7 and Figure 8), and no differences were observed between the untreated control (moMΦ) and moM(TGF-β) in the release of IL-6, IL-12, and TNF in response to Poly I:C stimulation (Figure 8).

Treatment with dexamethasone reduced the ability of macrophages to release proinflammatory cytokines in response to external stimuli. The moM(dexamethasone) presented a reduced ability to release IL-12 compared to the untreated control (moMΦ) in response to either MagPam2Cys_P80 or Poly I:C, with statistical significance (Figure 7 and Figure 8). In addition, treatment with this glucocorticoid reduced the release of IL-1α and IL-6 in response to stimulation with MagPam2Cys_P80, although IL-6 without statistical significance (Figure 8). Interestingly, moM(dexamethasone) released statistically significant lower levels of TNF compared to the untreated control (moMΦ) in response to MagPam2Cys_P80, but not Poly I:C (Figure 7 and Figure 8).

Very weak release of IL-18 was detected in response to both TLR agonists and we did not detect statistically significant differences between macrophage subsets (Appendix A).

We observed higher levels of IL-10 culture supernatant of the moM(IL-10) subset (Figure 6). Levels detected were extremely low (<0.2 ng/mL); thus, we might speculate that was not the result of de novo synthesis of this cytokine, but rather a residual of the original quantity added at time 0 (20 ng/mL). Similar levels were also observed in culture supernatants of moM(IL-10) after stimulation with either MagPam2Cys_P80 or Poly I:C. The moM1 presented enhanced ability to release low levels of IL-10 in response to TLR2 stimulation compared to the untreated control (moMΦ). On the contrary, moM(IL-4) and moM(TGF-β) both released lower levels of this immunosuppressive cytokine compared to the untreated control, in response to either the TLR2 or TLR3 agonist. Stimulation with dexamethasone resulted in decreased levels of IL-10 in culture supernatants compared to the untreated control (moMΦ) following stimulation with Poly I:C (Figure 7 and Figure 8). 

Higher levels of the receptor antagonist IL-1Ra were detected in culture supernatants of macrophages stimulated with IFN-γ + LPS (moM1) compared to the untreated control (moMΦ) in the absence of subsequent stimulation (Figure 7). This was expected, because stimulation with IFN-γ + LPS triggered release of this cytokine (Figure 6), and its release likely continued beyond 24 h post-stimulation. Stimulation with MagPam2Cys_P80 triggered little release of IL-1Ra, which was reduced by pretreatment with IL-4 or IL-10. Instead, stimulation with Poly I:C promoted a substantial release of IL-1Ra, which was reduced, with statistical significance, by pretreatment with all the tested polarizing factors (IFN-γ + LPS, IL-4, IL-10, and dexamethasone), except for TGF-β (Figure 7 and Figure 8). 

## 3. Discussion

Macrophages are a heterogeneous family of cells which are characterized by remarkable plasticity and versatility and are capable of responding to different microenvironmental signals by quickly modifying their phenotype and function [4]. Diverse macrophage subsets can either orchestrate or counteract inflammation [4]. Despite the increasing importance of pig biomedical models, very few studies have investigated macrophage polarization in this species. Previous studies have reported that classically activated porcine macrophages are characterized by enhanced expression of MHC class I and II molecules, activation markers (CD25), and co-stimulatory molecules [22], whereas few studies have investigated the impact of IL-4 or others “M2-related” polarizing factors in pigs [20,22,23]. In this study, we aimed to provide a deeper portrait of the phenotypic and functional changes of porcine moMΦ triggered by either IFN-γ + LPS (classical activation) or by diverse “M2-related” polarizing factors: IL-4, IL-10, TGF-β, or dexamethasone. Microscopy and flow cytometry were employed to analyze the effects of these five polarizing factors on moMΦ shape. We observed that exposure to IFN-γ + LPS resulted in slightly enhanced formation of cell clusters, as previously described [22], and we reported that neither IL-4, IL-10, TGF-β, nor dexamethasone altered moMΦ dimension or granularity, in agreement with our previous studies [19,20]. Singleton and colleagues reported that stimulation with IL-4 increased the numbers of elongated projections in macrophages [22], although we were unable to appreciate them in our study. 

Flow cytometry was employed to analyze the effects of the diverse stimuli on the expression of six surface markers. MHC class I and II DR expression was investigated since this can influence antigen presentation. MHC I and MHC II DR were upregulated by stimulation with IFN-γ + LPS, but not IL-4, in agreement with results previously published in pigs [19,23]. MHC II DR expression was downregulated by stimulation with IL-10, TGF-β, or dexamethasone, in line with the immune-suppressive activities of these molecules. CD14 is the receptor for LPS, and it is involved in clearance of Gram-negative bacteria [24]. We observed that this marker was downregulated by IL-4 (in agreement with that observed by Garcia-Nicolas and colleagues (2014) [23]), IL-10, or TGF-β stimulation, in agreement with our previously published work [20]. In this work, we observed that dexamethasone substantially upregulated CD14 expression, which contrasted with observations that have been described in humans, where researchers have observed that this glucocorticoid downregulated surface levels of CD14 on the human-transformed cell line THP-1 (a leukemia monocytic cell line) [25]. Further studies should investigate this peculiarity of pigs and whether higher doses of this glucocorticoid might have different impacts on this glycoprotein expression. CD16 is a low-affinity receptor for the IgG Fc, which facilitates antibody opsonization and antibody-dependent cellular cytotoxicity [26]. Our data revealed that stimulation with IL-10 and dexamethasone, but not TGF-β or IL-4, resulted in enhanced expression of CD16, in agreement with other previous publications in pigs [20,23]. Human macrophages exposed to IL-10 similarly presented enhanced expression of these markers compared to untreated macrophages or those exposed to IFN-γ + LPS [27,28]. CD163 is a scavenger receptor and it is often associated with anti-inflammatory macrophage phenotype [29]. In pigs, it has been reported that IL-4 stimulation triggered CD163 downregulation on macrophages [23], and we also observed little decrease in the expression of this scavenger receptor in moM(IL-4) compared to the untreated control (moMΦ), although without statistical significance. We observed that stimulation with IL-10, but not TGF-β, resulted in enhanced expression of this scavenger receptor, in agreement with our previous publication [20]. Porcine moMΦ treated with dexamethasone also presented increased CD163 expression, similarly to observations reported in pig monocytes and derived macrophages [22,30] and the immortalized porcine macrophage cell line IPKM [31]. As stated above, human M2 macrophages are characterized by high level expression of this scavenger receptor [32], but differences between subsets have been observed; it has been described that stimulation with IL-10 or dexamethasone, but not TGF-β, enhanced surface expression of this marker on human macrophages [33,34,35], similar to our observations in pigs. CD169 (SIGLEC1) contributes to antigen presentation and lymphocyte activation [36,37]. We observed that CD169 expression was significantly enhanced only after stimulation with IFN-γ + LPS. This is line with descriptions in humans and rodents, where CD169 upregulation on macrophages has been achieved by stimulation with either type I or type II IFNs [37]. In humans, it has been described that glucocorticoids could increase the expression of CD169 [7], and in pigs, Singleton et al. (2018) observed that dexamethasone enhanced the surface levels of this molecule on monocytes, although at notably higher doses than used in this study [30]. 

We further assessed the immunomodulatory effects of IFN-γ + LPS, IL-4, IL-10, TGF-β, or dexamethasone on porcine moMΦ through gene expression studies. The expression of 84 cytokine genes, including several proinflammatory or anti-inflammatory interleukins, chemokines, interferons, and tumor necrosis factor family members, were evaluated using PCR arrays 24 h post-stimulation. Expression of selected genes was also monitored over time with RT-PCR, as well as release of key immune cytokines through ELISA. As expected, classical activation enhanced expression and release of several proinflammatory cytokines; elevated levels of IL-1β, IL-6, IL-12, IL-23, and TNF are indeed regarded as a hallmark of M1 polarization in humans and mice [32,38]. Increased expression of several chemokines, CCL2, CCL4, CCL5, CCL8, CCL20, CCL23, CXCL8, and CXCL10 was observed, which reflected the proinflammatory phenotype of these cells. Only a few genes were downregulated in moM1 compared to the untreated control with *p*-value < 0.1: CCL21 and TGF-β2. TGF-β2 is one of the three isoforms of TGF-β [39] and in humans it has been observed that IFN-γ reduced both basal- and IL-4-stimulated release of TGF-β2 by bronchial epithelial cells [40]. However, CCL21 downregulation was unexpected, since in humans this chemokine has promoted chemotaxis of M1 but not M2 macrophages [41]. CCL21 downregulation might be a protective mechanism as it may limit recruitment of M1 in the inflammatory sites, preventing exacerbated and pathological inflammation. IL-4 stimulation of macrophages gave rise to a different phenotype, characterized by significant (*p* < 0.05) upregulation of just two cytokine genes: BMP2 and IL-18. Bone morphogenetic protein 2 (BMP-2) is a member of the TGF-β superfamily and it plays an important role in the development of bone and cartilage [42]. Enhanced levels of this cytokines are in line with “M2 polarization”, which is associated with osteogenesis and promotion of bone mineralization [43]. In addition, in mice, it has been described that BMP-2 decreased expression of M1 phenotypic markers, such as IL-1β, IL-6, and iNOS, in M1-polarized macrophages, whereas it enhanced expression of the enzyme Arginase 1 (Arg-1), suggesting this protein may shift macrophages to M2-like phenotypes [44]. IL-18 is a member of the IL-1 superfamily and a potent inducer of IFN-γ; it is a proinflammatory, but not pyrogenic, cytokine. It synergizes with IL-12 to activate NK cells and cytotoxic T cells [45], but it has been described that it can enhance other T-cell responses, such as Th17 cells, in synergy with IL-23 or Th2 responses [46]. In humans, classical (M1) and not alternative (M2) activation triggers upregulation of this proinflammatory cytokine [32], whereas in pigs we observed that IL-4 and IL-10 both enhanced its expression. However, increased IL-18 gene expression in response to IL-4 or IL-10 treatments was not associated with enhanced IL-18 protein levels in culture supernatants of moM(IL-4) or moM(IL-10) compared to the untreated control (moMΦ). This suggests that factors at a post-transcriptional level counteract the release of this cytokine. In humans and rodents, it has been described that activation with IL-4 was characterized by enhanced expressions of IL-10 and the chemokines CCL17 and CCL22, the latter two inhibited by IFN-γ [1,32,38]; however, in this study, we observed that IL-4 did not enhance expression of IL-10. ELISA data confirmed the absence of IL-10 release in response to IL-4 stimulation, whereas a small but statistically significant release was seen in culture supernatants of moM1. Array data revealed that IL-4 enhanced (fold changes > 2.0) CCL17 and CCL22 expressions, although without statistical significance. In addition, stimulation with IFN-γ + LPS also resulted in enhanced expression of both chemokine genes. Our data highlighted interesting peculiarities of this species and suggest that neither IL-10, CCL17, nor CCL22 can be used as hallmarks of M(IL-4) polarization in pigs.

As stated above, IL-10 is regarded as a potent immune-suppressive cytokine, which limits production of proinflammatory interleukins, chemokines, and TNF (formerly known as TNF-α) [47]. In line with this immunosuppressive phenotype, our array data revealed that IL-10 stimulation promoted downregulation of several proinflammatory cytokines, and triggered significant upregulation of a few cytokine genes, including IL-18. We unexpectedly observed upregulation of this proinflammatory IL-1 family member 24 h post-stimulation with IL-10, similar to that observed in moM(IL-4), although no enhanced levels of IL-18 protein were observed in culture supernatants. This is an interesting peculiarity of pigs, and future studies should better investigate whether alternative macrophage activation in this species is characterized by induction of IL-18 and not IL-10, which is the opposite of that observed in humans and mice [32,47]. 

Exposure to TGF-β resulted in downregulation of several proinflammatory cytokines, in line with the immunosuppressive action of these molecules on macrophages described either in humans [48] or in pigs [20]. Array data showed that only one cytokine gene was upregulated with *p*-value < 0.05: TGF-β2. TGF-β2 is a member of the TGFβ superfamily [39] and it is characterized by anti-inflammatory activity [49]. Accordingly, its enhancement reflects the immunosuppressive phenotype of moM(TGF-β). 

Glucocorticoids are drugs that have been developed to switch inflammation off [7]; thus, it was not unexpected to observed that stimulation with dexamethasone gave rise to a macrophage phenotype characterized by downregulation (fold change > 2) of 32 out of 84 tested cytokine genes. Only one gene was upregulated with statistical significance (*p* < 0.05): CCL23. CCL23 is a chemokine with immunosuppressive activity that, in humans, inhibits myeloid progenitor cell development and promotes selective recruitment resting T lymphocytes and not activated T lymphocyte monocytes [50,51]. Although this is in line with the anti-inflammatory phenotype of moM(dexamethasone), it was interesting to observe that none of the other tested “M2-related” polarizing molecules enhanced CCL23 expression. In humans, instead, it has been reported that IL-4 and IL-13 could both induce CCL23 production by monocytes [52]. In addition, our RT-PCR data showed that CCL23 was upregulated following IFN-γ + LPS stimulation. These results further emphasized the heterogeneity of the macrophage family and revealed further species differences.

In humans and mice, stimulation of macrophages with IL-10, TGF-β, and glucocorticoids are associated with enhanced expression and release of IL-10 [32], but we did not observe this in pigs. These data agree with our previous studies on IL-10 and TGF-β [20], and here, we expanded our observation to dexamethasone. Thus, we tested induction and release of another potent immunosuppressive cytokine: IL-1Ra. IL-1Ra is a receptor antagonist. It binds IL-1R1 with higher affinity than that of IL-1α or IL-1β, but without activation of the IL-1 signaling and the subsequent activation of inflammatory responses [45,53]. High levels of IL-1Ra were released following stimulation with IFN-γ + LPS, and it could be speculated that this was a protective mechanism developed by macrophages. MoM1 are characterized by elevated release of IL-1α, IL-1β, and other proinflammatory cytokines (IL-6, CXCL8, and IL-12); thus, IL-1Ra is likely released to counteract their activity, in order to avoid pathogenic inflammatory responses. We observed that IL-1Ra was only modestly expressed and released by moM(IL-4) compared to the untreated control, and none of the tested immunosuppressive molecules enhanced its release. Stimulation with IL-10, but not TGF-β or dexamethasone, promoted its expression over time. In other species, in contrast, IL-1Ra is associated with alternative (IL-4) and not classical activation of macrophages [1,32,54]. It is interesting to observe that stimulation of porcine macrophages with IL-4 induced only a little induction/release of IL-1Ra, which was sustained in moM1, but instead promoted expression of another IL-1 family member: IL-18. Future studies should better understand factors underling this peculiarity of pigs and whether it is extended to other members of the IL-1 family. 

In the final part of the study, we investigated the functionality of the different macrophage subsets generated by exposure to diverse stimuli. TLRs are a family of pattern recognition receptors that recognize pathogen-associated molecular patterns (PAMPs), with subsequent activation of signaling cascades which culminate in inflammasome activation, and consequent inflammatory responses [55,56]. In this study, we investigated the ability of the six diverse macrophage subsets to release proinflammatory cytokines in response to either a TLR2 ligand (MagPam2Cys_P80) or a TLR3 ligand (Poly I:C). In our previous studies, we observed that moM(IL-10) and moM(TGF-β) differed in their ability to release proinflammatory cytokines in response to both the TLR2 and the TLR4 agonist stimulation; proinflammatory cytokine release was drastically impaired by IL-10, but to a much lower extent by TGF-β [20]. Although differences between tested animals were observed, our data revealed that IL-4 presented only a limited impact on the macrophage’s ability to respond to external stimuli, whereas moM(IL-10) presented a marked anti-inflammatory phenotype, with reduced ability to release proinflammatory cytokines in response to either MagPam2Cys or Poly I:C stimulation, in agreement with our previous work [20]. MoM(TGF-β) presented a less marked anti-inflammatory phenotype compared to moM(IL-10): exposure to TGF-β did not statistically significantly impair the ability of moMΦs to release IL-1α and IL-1β in response to MagPam2Cys_P80 lipopeptide, in agreement with our previous work [20], and it downregulated IL-12 release in response to the tested TLR ligand with less intensity compared to IL-10 or dexamethasone. These differences are in line with the pleiotropic nature of TGF-β that possesses regulatory and inflammatory activities (in the presence of IL-6, this cytokine can indeed drive the differentiation of Th17 cells, further promoting inflammation) [57]. In this work, the ability of dexamethasone to impair porcine macrophage response to either TLR2 or TLR3 ligands was also analyzed, and we observed that this glucocorticoid presented a reduced ability of macrophages to release proinflammatory cytokines in response to the tested PAMPs in a similar manner compared to IL-10. These data are in line with the anti-inflammatory activity of these types of molecules. Finally, the release of anti-inflammatory cytokines was tested. Although we observed differences between the three tested blood donor pigs, our data revealed that neither MagPam2Cys_p80 nor Poly I:C promoted release of IL-10, as expected, whereas Poly I:C induced enhanced release of IL-1Ra from macrophages. This is in line with results described in humans and mice, where researchers have observed that stimulation with Poly I:C activated TLR3, with subsequent intracellular signaling that resulted in activation of transcription factors IRF3 and NF-κβ, triggering enhanced expression of the receptor antagonist IL-1Ra [58]. We observed that either classical activation (IFN-γ and LPS) or “M2 polarizing factors” decreased TLR3-mediated IL-1Ra release, with the exception of TGF-β. It has been described that TGF-β promoted the induction of IL-1Ra, likely in an IL-1 dependent manner [59]; thus, it was perhaps not unexpected that IL-1Ra release from TLR3 stimulated porcine moM(TGF-β) was unaffected.

Overall, we observed differences between stimulation with IFN-γ + LPS (M1) and “M2-related” factors, and also between immunosuppressive molecules, such as IL-10, TGF, and dexamethasone. Our data also suggest it would be more appropriate to apply nomenclature linked to the activator(s) used, such as M(IL-10), M(IL-10), M(TGF-β), M(dexamethasone), as suggested by Murray et al. (2014) [8], to porcine macrophages.

## 4. Materials and Methods

### 4.1. Animals and Ethical Statement 

Six cross-bred pigs (*Sus scrofa domesticus*) of either sex, aged 6–18 months old, were used as blood donors for in vitro experiments. Pigs were housed at the Experimental Station of Istituto Zooprofilattico Sperimentale (IZS) of Sardinia (“Surigheddu”, Sassari, Italy). Animal husbandry, handling, and procedures (bleeding) were carried out according to the Italian Legislative Decree No. 26 dated 4 March 2014 and in agreement with the Guide of Use of Laboratory Animals issued by the Italian Ministry of Health (authorization No. 1232/2020-PR). 

Heparinized blood samples were used for generation of monocyte-derived macrophages (moMΦ) (described in Section 4.2). Animal health was routinely monitored by trained veterinarians, and blood samples were screened for several porcine pathogens. The absence of African swine fever (ASFV), porcine parvovirus (PPV), and porcine circovirus 2 (PCV2) genome was evaluated though qualitative real-time PCR, as previously described [21,60], with primers reported in the Appendix A [61,62,63]. The absence of the porcine reproductive and respiratory syndrome virus (PRRSV) and *Mycoplasma hyopneumoniae* was monitored using commercial real-time PCR kits (LSI VetMAX™ PRRSV EU/NA and VetMAX™-Plus qPCR Master Mix, both Thermo Fisher Scientific, respectively), following the manufacturer’s instructions [21].

### 4.2. Generation of Porcine Monocyte-Derived Macrophages and Stimulation with Diverse Polarizing Factors 

Monocyte-derived macrophage (moMΦ) cultures were obtained from blood leukocytes using Petri dishes and through the addition of 50 ng/mL of recombinant human M-CSF (hM-CSF) (Thermo Fisher Scientific, Waltham, MA, USA) to the culture media (RPMI-1640 supplemented with 10% fetal bovine serum (FBS), 100 U/mL penicillin, and 100 μg/mL streptomycin (complete RPMI, cRPMI), as we previously described [21,64,65]. The moMΦ cells were seeded in 12-well plates (Greiner CELLSTAR, Sigma-Aldrich, Saint Louis, MO, USA) (1 × 10^6^ live cells per well) or 4-well chamber slides (Nunc Lab-Tek chamber slide system, Thermo Fisher Scientific) (3 × 10^5^ live cells per well). After seeding, cells were cultured in unsupplemented fresh cRPMI at 37 °C, 5% CO_2_, then 24 h later, the moMΦs were left untreated, or they were stimulated for 24 h with several polarizing factors. 

The moM1 cells were generated using recombinant porcine IFN-γ (Raybiotech Inc, Norcross, GA, USA) and LPS (lipopolysaccharide from *Escherichia coli* 0111:B, Sigma-Aldrich), both at 100 ng/mL [19,22,34,65]. Other monocyte-derived macrophage subsets were generated though supplementation of the culture media with “M2-related” polarizing factors, recombinant porcine IL-4, IL-10, TGF-β (all R&D Systems, Minneapolis, MN, USA) [19,20,65], or dexamethasone (Sigma-Aldrich), all at 20 ng/mL. 

### 4.3. Assessment of Cell Morphology

Cell morphology was investigated on macrophage subsets seeded in 4-well chamber slides, 24 h post-stimulation, by either fluorescence or phase-contrast microscopy. For fluorescence microscopy, macrophages were fixed with 4% paraformaldehyde, washed with PBS, and subsequently labeled with Alexa Fluor 488 conjugated phalloidin and Hoechst (both Molecular Probes, Thermo Fisher Scientific, Rockford, IL, USA) to visualize actin cytoskeleton or nuclei, respectively [20]. Microscopy was carried out using an inverted stereo microscope (Olympus IX 70, Segrate, Italy) with magnification 40× objective and processed with the LAS AF Lite software 1.0.0(Leica Microsystem, Wetzlar, Germany), as previously reported [20]. For light microscopy, macrophage subsets were fixed with 4% paraformaldehyde, washed with PBS, and phase-contrast images were acquired using an inverted microscope (Olympus IX70, Segrate, Italy) equipped with a 20×/0.40 numeric aperture objective lens [21].

### 4.4. Flow Cytometry

Flow cytometry was performed to determine the expression of cell surface markers, as well as dimension and granularity, as previously published [20,21]. In detail, the moMΦ were seeded in 12-well plates, and then they were stimulated (see Section 4.2). Then, the cells were harvested with 10 mM EDTA in PBS and transferred to 5 mL round bottom tubes (Corning, Corning, NY, USA). Cells were first stained with Zombie Aqua viability dye (BioLegend, San Diego, CA, USA) (30 min, room temperature), then they were washed with PBS supplemented with 0.5% bovine serum albumin (BSA), and subsequently stained with several murine monoclonal antibodies (mAbs): anti-porcine CD16-PE (clone G7, Thermo Scientific Pierce, Rockford, IL, USA), anti-human CD14-PerCP-Cy5.5 (clone Tuk4, Miltenyi Biotec, Bergisch Gladbach, Germany) [66], CD163-PE (clone 2A10/11, Bio-Rad Antibodies, Kidlington, UK), CD169-FITC (clone 3B11/11, Bio-Rad Antibodies), anti-pig MHC I (clone JM1E3, Bio-Rad Antibodies), and anti-pig MHC II DR (clone 2E9/13, Bio-Rad Antibodies) (Appendix A). MHC I and MHC II DR expressions were visualized by subsequent staining with BV421 rat anti-mouse IgG1 (clone A85-1, BD Horizon BD Biosciences, Franklin Lakes, NJ, USA) or BV786 rat anti-mouse IgG2b (clone R12-3, BD Horizon BD Biosciences), respectively. All mAbs were incubated with cells for 15 min at 4 °C, cells were washed with PBS supplemented with 2% FBS, and resuspended in PBS supplemented with 2 mM EDTA. 

Analysis was carried out using a FACS Celesta flow cytometer (BD Biosciences), acquiring 5000 live moMΦs. Data analyses were performed using the BD FACS Diva Software 8.0 (BD Biosciences), by exclusion of doublets, gating on viable moMΦ, and then assessing the staining for surface markers [20,21].

### 4.5. Cytokine Release in Response to Stimulation 

Monocyte-derived macrophages (moMΦ) were left untreated or they were stimulated with diverse polarizing factors: IFN-γ + LPS, IL-4, IL-10, TGF-β, or dexamethasone (as described in Section 4.2). Then 24 h later, cytokine contents in culture supernatants were determined using multiplex ELISA, as previously described [21,64,65]. In brief, culture supernatants were removed, centrifuged at 2000× *g* for 3 min to remove cell debris, and stored at −80 °C until analyzed. Levels of IL-1α, IL-1β, IL-1Ra, IL-6, CXCL8, IL-10, IL-12, IL-18, and TNF were quantified using the Porcine Cytokine/Chemokine Magnetic Bead Panel Multiplex assay (Merck Millipore, Darmstadt, Germany) and a Bioplex MAGPIX Multiplex Reader (Bio-Rad, Hercules, CA, USA), according to the manufacturers’ instructions. 

### 4.6. Impact of Diverse Polarizing Factors on Key Immune Cytokine Gene Expression

The moMΦ cells were seeded in 12-well plates, and then left untreated or they were stimulated with diverse polarizing factors: IFN-γ and LPS, IL-4, IL-10, TGF-β, or dexamethasone (as described in Section 4.2). Then, 4, 8, and 24 h later, cells were harvested to evaluate gene expression of selected cytokines and TLRs. 

The RNeasy Mini Kit (QIAGEN, Hilden, Germany) was employed to extract total RNA, which was eluted in 50 µL of ultrapure RNase-free water. 250 ng of the obtained purified RNA was used as the template for cDNA synthesis, as previously described [21]. Subsequently, RT-qPCR was employed to determine the expressions of several cytokine genes (*IL-1β, IL-1RA, IL-10, IL-18, TGF-β2, CXCL13, CCL23, IL-6, TNF, TLR2,* and *TLR3*), using the primer sets listed in the Appendix A [67,68,69,70,71,72]. 

For all tested genes, five independent experiments using different blood donor animals were performed. In each sample, the relative gene expression levels were calculated from Cq (quantification cycle) values using the classical and widely adopted 2^−∆∆Cq^ method [21,70,73].

### 4.7. RNA Extraction and PCR Array Analysis

PCR arrays for 84 genes related to pig cytokines and chemokines were measured on macrophage subsets generated using three blood donor pigs. For each animal, six macrophage subsets were obtained: moMΦ, moM1 (IFN-γ + LPS), moM(IL-4), moM(IL-10), moM(TGF-β), and moM(dexamethasone). RNA was extracted from the cell monolayers using an miRNAeasy Mini Kit (QIAGEN). Genomic DNA was digested using an RNase-Free DNase set (QIAGEN). The concentration of RNA was determined using a Qubit 4 fluorometer (Thermo Fisher). Total RNA (500 ng) was used for cDNA synthesis using a RT2 First Strand Kit (QIAGEN). The RNA quality was assessed by an RT2 RNA QC PCR Array (QIAGEN). Real-time PCR was then conducted using an RT2 Profiler PCR Array for pig cytokines and chemokines (QIAGEN, Cat. No. 330231 PASS-150ZC). The data analysis was performed using the GeneGlobe Data Analysis Center available at QIAGEN (https://geneglobe.qiagen.com/us/analyze, accessed on 25 January 2023). A list of genes is shown in Appendix A (according to information provided by the manufacturer). All data were normalized to an average of five housekeeping genes (ACTB, B2M, GAPDH, HPRT1, and RPLP0) (Appendix A). The relative gene expression levels compared to the untreated control were then calculated using the classical and widely adopted ∆Ct method (2^−∆∆Ct^) [73]. Unsupervised hierarchical clustering was performed to indicate the co-regulated genes across groups.

### 4.8. Stimulation with TLR2 or TLR3 Agonists

The moMΦ cells were seeded in 12-well plates, and then they were left untreated, or they were stimulated with diverse polarizing factors: IFN-γ and LPS, IL-4, IL-10, TGF-β, or dexamethasone (as described in Section 4.2). Then 24 h later, the culture media was removed and replaced with cRPMI supplemented with either a TLR2 agonist (S-[2–bis(palmitoyl)-propyl]cysteine (Pam2Cys) lipopeptide, 100 ng/mL, Espikem, Prato, Italy [21,70]) or a TLR3 agonist (poly I:C, 100 ng/mL, Sigma-Aldrich). Then, 24 h post-stimulation, culture supernatants were removed, centrifuged at 2000× *g* for 3 min (to remove cell debris), and stored at −80 °C until determination of cytokine levels, as described in Section 4.5. 

### 4.9. Statistical Analysis 

In vitro experiments were performed in technical duplicate and repeated with at least three different blood donor pigs. 

Data were first checked for normality using the Shapiro–Wilk test, then they were graphically and statistically analyzed with GraphPad Prism 9.01 (GraphPad Software Inc., La Jolla, CA, USA). 

Flow cytometry, ELISA, and qPCR data were presented as box-and-whisker plots, showing the median and interquartile range (boxes) and minimum and maximum values (whiskers). These data were analyzed using either the parametric unpaired *t*-test or the nonparametric Mann–Whitney test; *p*-values lower than 0.05 were considered to be statistically significant (* *p* < 0.05, ** *p* < 0.01, and *** *p* < 0.001).

The PCR array data were presented as a heatmap. PCR array for 84 genes were analyzed using the GeneGlobe Data Analysis Center available at QIAGEN (https://geneglobe.qiagen.com/us/analyze, accessed on 25 January 2023), as described in Section 4.7. Student’s *t*-tests were employed to evaluate statistical differences, and a statistically significant difference was set as *p* < 0.05.

## 5. Conclusions

Overall, our data highlighted the remarkable heterogeneity and plasticity of porcine macrophages and showed that even molecules with similar biological functions (IL-10, TGF-β, dexamethasone) gave rise to distinct phenotypes. In addition, some porcine-specific peculiarities were observed, such as no induction or release of IL-10 in response to any of the four “M2-related” polarizing factors tested. In addition, IL-4 and IL-10, unexpectedly, both enhanced expression of proinflammatory IL-18, although this did not translate to increased secretion of this cytokine. Information generated by this study can help researchers to better interpret in vitro and in vivo results of host–pathogen interaction studies and will benefit researchers using pigs as a biomedical model. 

## Figures and Tables

**Figure 1 ijms-24-04671-f001:**
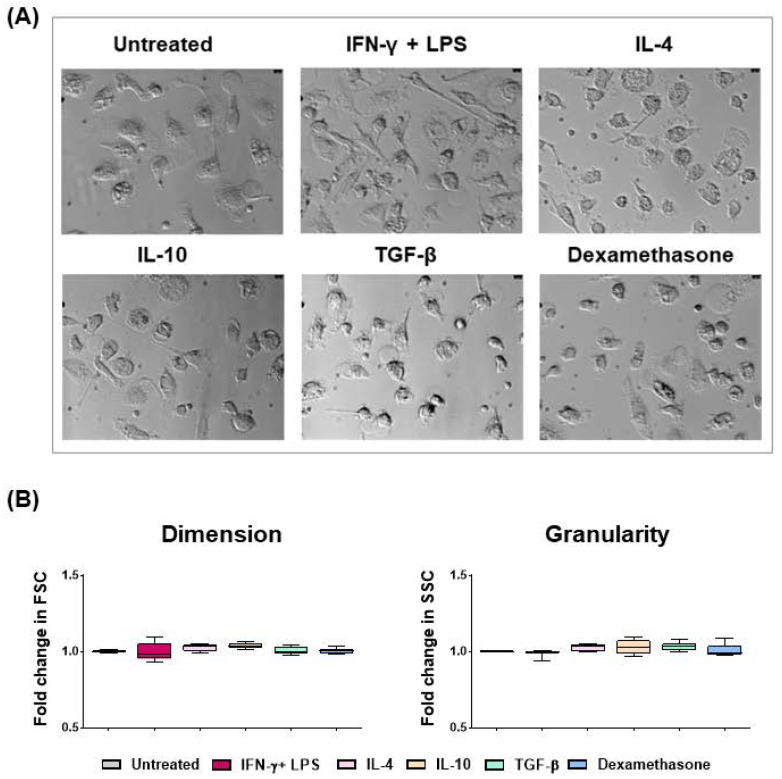
Porcine monocyte-derived macrophage subsets morphology. Porcine moMΦ were left untreated, or they were stimulated with diverse polarizing factors: IFN-γ + LPS (both at 100 ng/mL), IL-4 (20 ng/mL), IL-10 (20 ng/mL), TGF-β (20 ng/mL), or dexamethasone (20 ng/mL). Then, 24 h post-stimulation, morphologies were evaluated using phase contrast or fluorescence microscopy, as well as flow cytometry: (**A**) Phase contrast microscopy images were acquired using an inverted microscope, with a magnification 20×. Scale bar, 10 μm. Images of six representative macrophage subsets, one from each condition (untreated, IFN- + LPS, IL-4, IL-10, TGF-β, and dexamethasone) are presented; (**B**) flow cytometry was employed to evaluate changes in the dimension and granularity of moMΦ. Forward scatter (FSC) and side scatter (SSC) data are presented as fold change relative to the untreated control (moMΦ). Mean data for quadruplicate biological replicates and standard deviation (SD) are presented. Values of treated macrophages were compared to the untreated control (moMΦ), using an unpaired *t*-test of a Mann–Whitney test.

**Figure 2 ijms-24-04671-f002:**
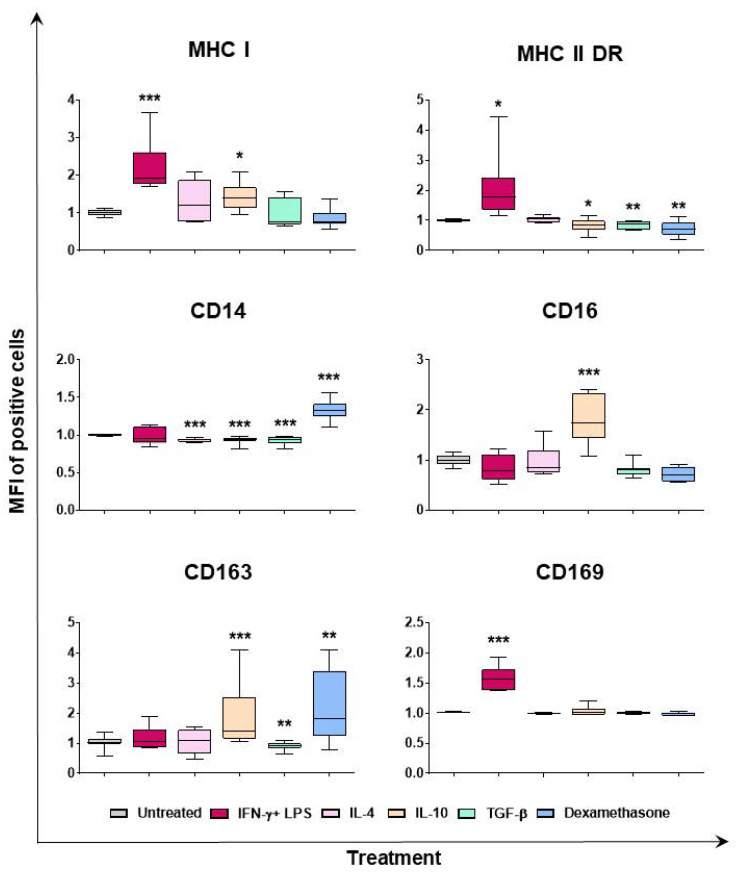
Effect of diverse polarizing factors on porcine moMΦ surface marker expressions (mean of fluorescence intensity). Porcine moMΦ were left untreated, or they were stimulated with diverse polarizing factors: IFN-γ + LPS (both at 100 ng/mL), IL-4 (20 ng/mL), IL-10 (20 ng/mL), TGF-β (20 ng/mL), or dexamethasone (20 ng/mL). Then, 24 h post-stimulation, flow cytometry was employed to determine the expression of several surface markers: MHC I, MHC II DR, CD14, CD16, CD163, and CD169. Mean fluorescence intensity (MFI) of positive cells was evaluated, and MFI data are expressed as fold change relative to the un-activated condition (moMΦ). Data from three independent experiments utilizing different blood donors are presented. Data are displayed as box-and-whisker plots, showing the median and interquartile range (boxes) and minimum and maximum values (whiskers). Values of treated macrophages were compared to the untreated control (moMΦ), using an unpaired *t*-test of a Mann–Whitney test. *** *p* < 0.001, ** *p* < 0.01, and * *p* < 0.05.

**Figure 3 ijms-24-04671-f003:**
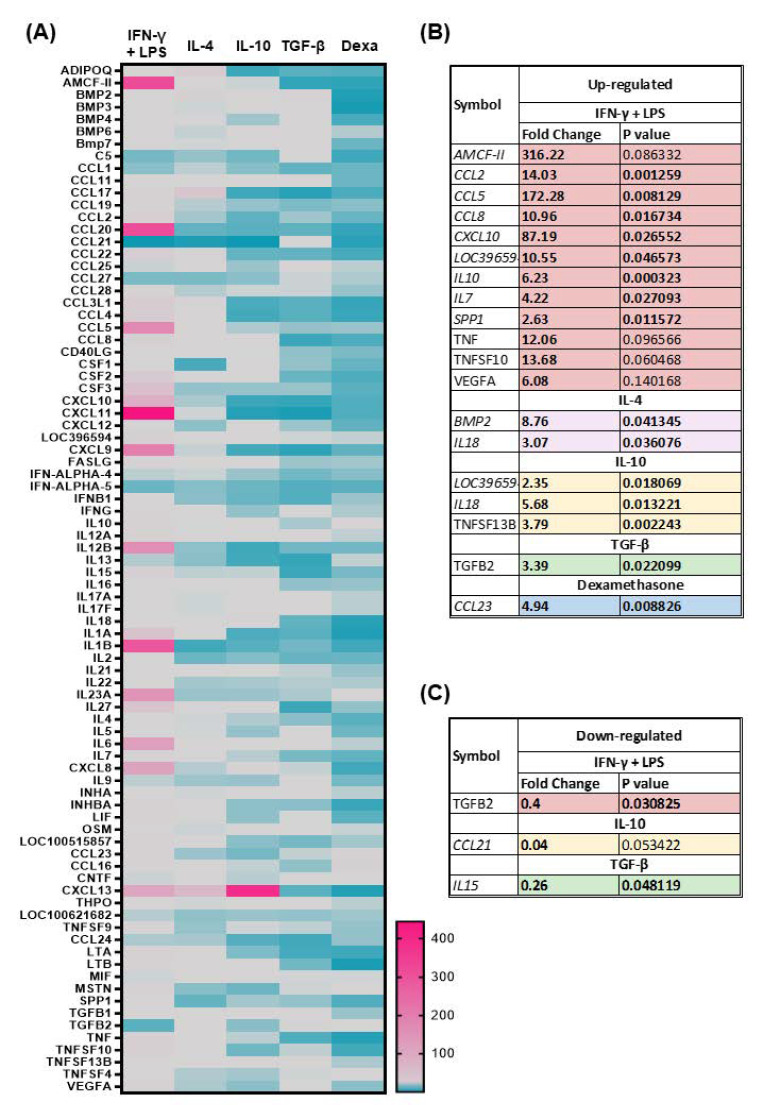
Modulation of 84 genes in moMΦ stimulated with diverse polarizing factors. Porcine moMΦ were left untreated, or they were stimulated with diverse polarizing factors: IFN-γ + LPS (both at 100 ng/mL), IL-4 (20 ng/mL), IL-10 (20 ng/mL), TGF-β (20 ng/mL), or dexamethasone (“DEXA”, 20 ng/mL). Then, 24 h post-stimulation, macrophage subsets were analyzed using the RT2 Profiler PCR Array for 84 common immune-related genes: (**A**) The heatmap illustrates fold change expression of these 84 immune-related genes, obtained from macrophage subsets from three diverse pig blood donors. For each macrophage subset, fold change in gene expression was calculated relative to the untreated control (moMΦ). The colors represent the fold change in gene expression compared to the untreated control, with the brightest pink representing the highest value, light grey representing the baseline value (fold change = 1), and blue representing the smallest value; (**B**) for each macrophage subsets, statistically significantly upregulated genes (fold change > 2, *p*-value < 0.05) are presented, with the corresponding fold change and *p*-value; (**C**) for each macrophage subsets, statistically significantly downregulated genes (fold change > 2, *p*-value < 0.05) are presented, with the corresponding fold change and *p*-value.

**Figure 4 ijms-24-04671-f004:**
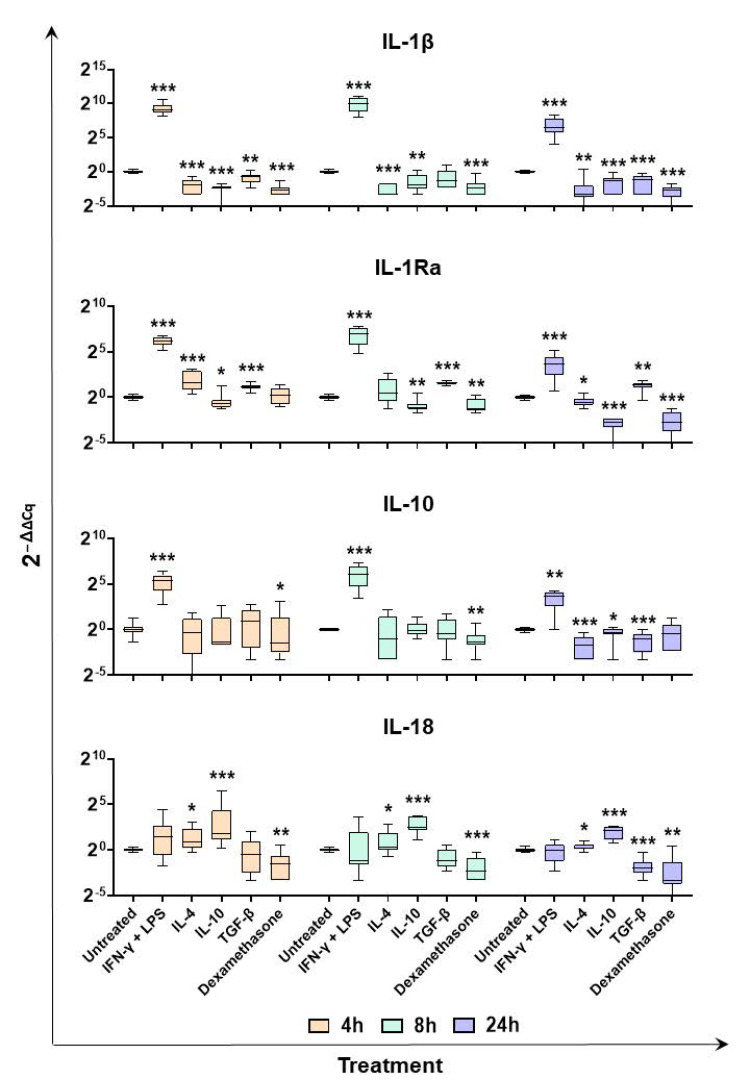
Expression of selected interleukin genes over time in moMΦ stimulated with diverse polarizing factors. Porcine moMΦ were left untreated, or they were stimulated with diverse polarizing factors: IFN-γ + LPS (both at 100 ng/mL), IL-4 (20 ng/mL), IL-10 (20 ng/mL), TGF-β (20 ng/mL), or dexamethasone (20 ng/mL). At 4, 8, and 24 h post-stimulation, gene expression levels of *IL-1β, IL-1Ra, IL-10,* and *IL-18* were determined using qPCR. At each timepoint, data were normalized to the values of the untreated control (moMΦ) and expressed as 2^−ΔΔCq^, where ΔCq = Cq (target gene) − Cq (house-keeping gene), and ΔΔCq = ΔCq (stimulated samples) − ΔCq (untreated samples). Data from five independent experiments utilizing different blood donors are presented. Data are displayed as box-and-whisker plots, showing the median and interquartile range (boxes) and minimum and maximum values (whiskers). Values of treated macrophages were compared to the untreated control (moMΦ), using an unpaired *t*-test of a Mann–Whitney test. *** *p* < 0.001, ** *p* < 0.01, and * *p* < 0.05.

**Figure 5 ijms-24-04671-f005:**
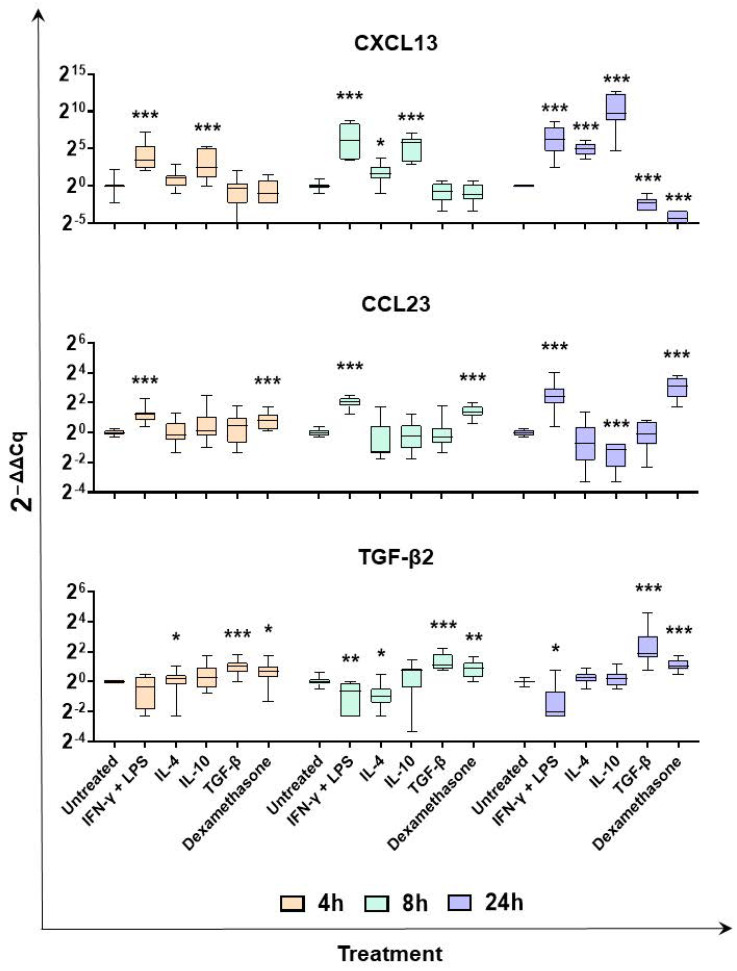
Expression of TGF-β2, CXCL13, and CCL23 over time in moMΦ stimulated with diverse polarizing factors. Porcine moMΦ were left untreated, or they were stimulated with diverse polarizing factors: IFN-γ + LPS (both at 100 ng/mL), IL-4 (20 ng/mL), IL-10 (20 ng/mL), TGF-β (20 ng/mL), or dexamethasone (20 ng/mL). At 4, 8, and 24 h post-stimulation, gene expression levels of *TGF-β2, CXCL13,* and *CCL23* were determined using qPCR. At each timepoint, data were normalized to the values of the untreated control (moMΦ) and expressed as 2^−ΔΔCq^, where ΔCq = Cq (target gene) − Cq (house-keeping gene), and ΔΔCq = ΔCq (stimulated samples) − ΔCq (untreated samples). Data from five independent experiments utilizing different blood donors are presented. Data are displayed as box-and-whisker plots, showing the median and interquartile range (boxes) and minimum and maximum values (whiskers). Values of treated macrophages were compared to the untreated control (moMΦ), using a Mann–Whitney test. *** *p* < 0.001, ** *p* < 0.01, and * *p* < 0.05.

**Figure 6 ijms-24-04671-f006:**
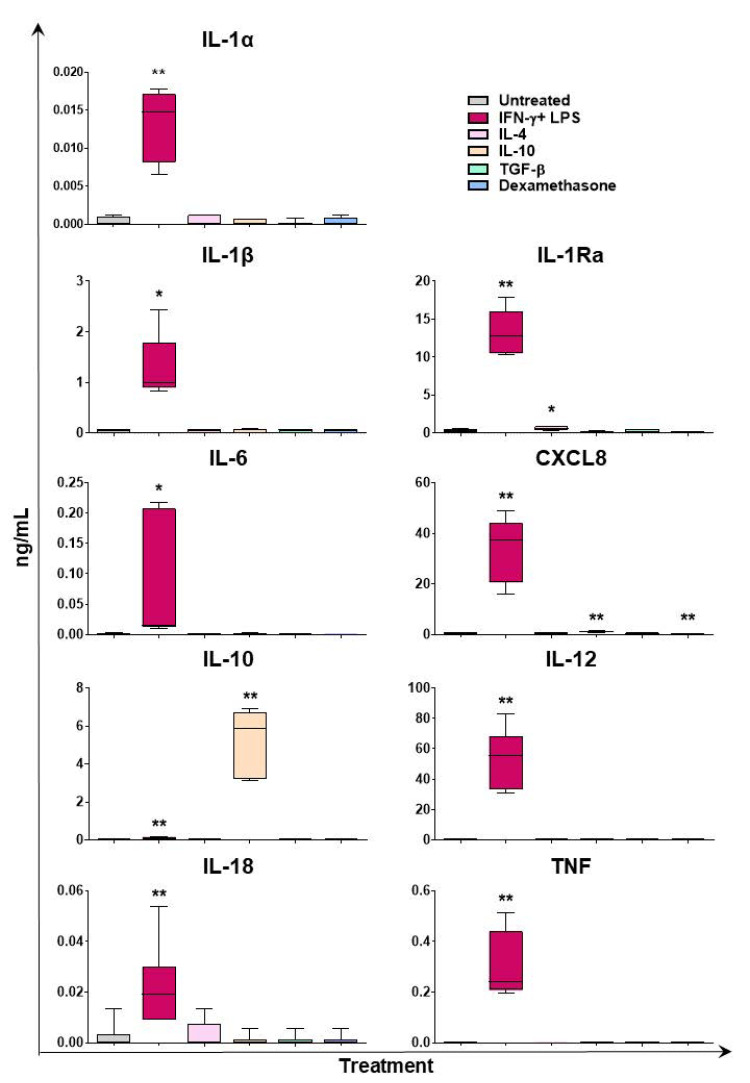
Cytokine content in culture supernatant of moMΦ stimulated with diverse polarizing factors. Porcine moMΦ were left untreated, or they were stimulated with diverse polarizing factors: IFN-γ + LPS (both at 100 ng/mL), IL-4 (20 ng/mL), IL-10 (20 ng/mL), TGF-β (20 ng/mL), or dexamethasone (20 ng/mL). Then, 24 h post-stimulation, levels of IL-1α, IL-1β, IL-1Ra, IL-6, CXCL8, IL-10, IL-12, IL-18, and TNF were determined using a multiplex ELISA. Data from three independent experiments utilizing different blood donors are presented. Data are displayed as box-and-whisker plots, showing the median and interquartile range (boxes) and minimum and maximum values (whiskers). Values of treated macrophages were compared to the untreated control (moMΦ) using a Mann–Whitney test. ** *p* < 0.01, and * *p* < 0.05.

**Figure 7 ijms-24-04671-f007:**
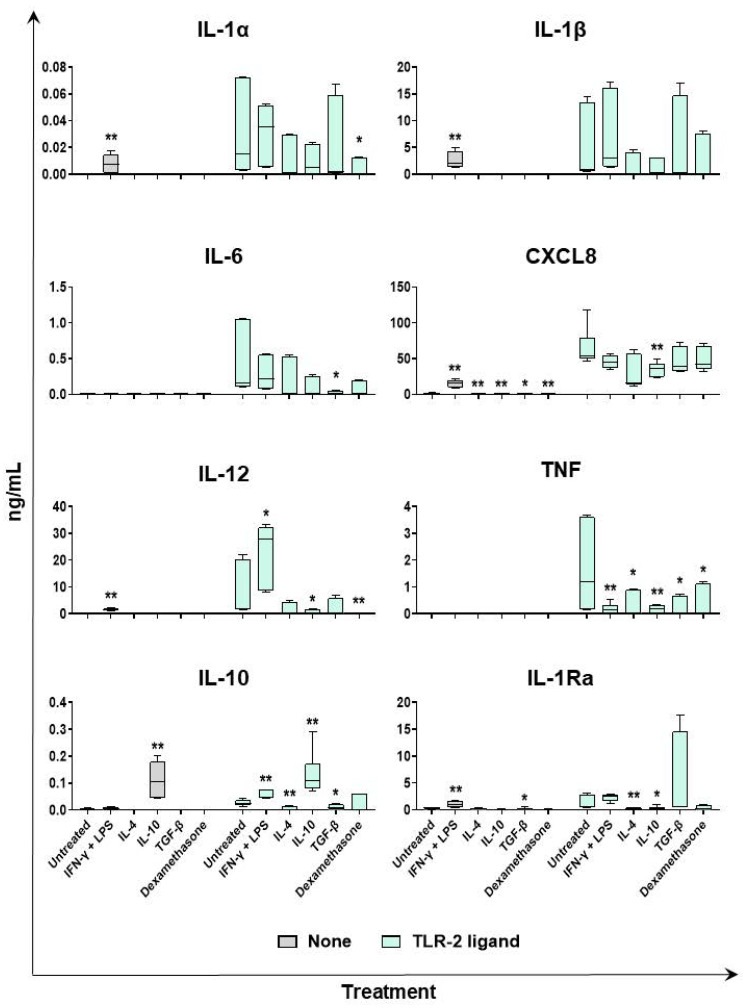
Ability of diverse macrophage subsets to release proinflammatory and anti-inflammatory cytokines in response to TLR2 agonist stimulation. The moMΦ were left untreated, or they were stimulated with diverse polarizing factors: IFN-γ + LPS (both at 100 ng/mL), IL-4 (20 ng/mL), IL-10 (20 ng/mL), TGF-β (20 ng/mL), or dexamethasone (20 ng/mL). Then 24 h later, culture supernatants were replaced with fresh media and cells were left untreated or activated using a TLR-2 ligand (Mag-Pam2Cys_P80, 100 ng/mL); 24 h later, the amounts of IL-1α, IL-1β, IL-6, CXCL8, IL-12, TNF, IL-1Ra, ad IL-10 in culture supernatants were determined using a multiplex ELISA. Data from three independent experiments utilizing different blood donors are presented. Data are displayed as box-and-whisker plots, showing the median and interquartile range (boxes) and minimum and maximum values (whiskers). For each cytokine (IL-α, IL-β, IL-6, CXCL8, IL-12, TNF, IL-10, and IL-1Ra), values of treated macrophages were compared to the untreated control (moMΦ) using a Mann–Whitney test. ** *p* < 0.01, and * *p* < 0.05.

**Figure 8 ijms-24-04671-f008:**
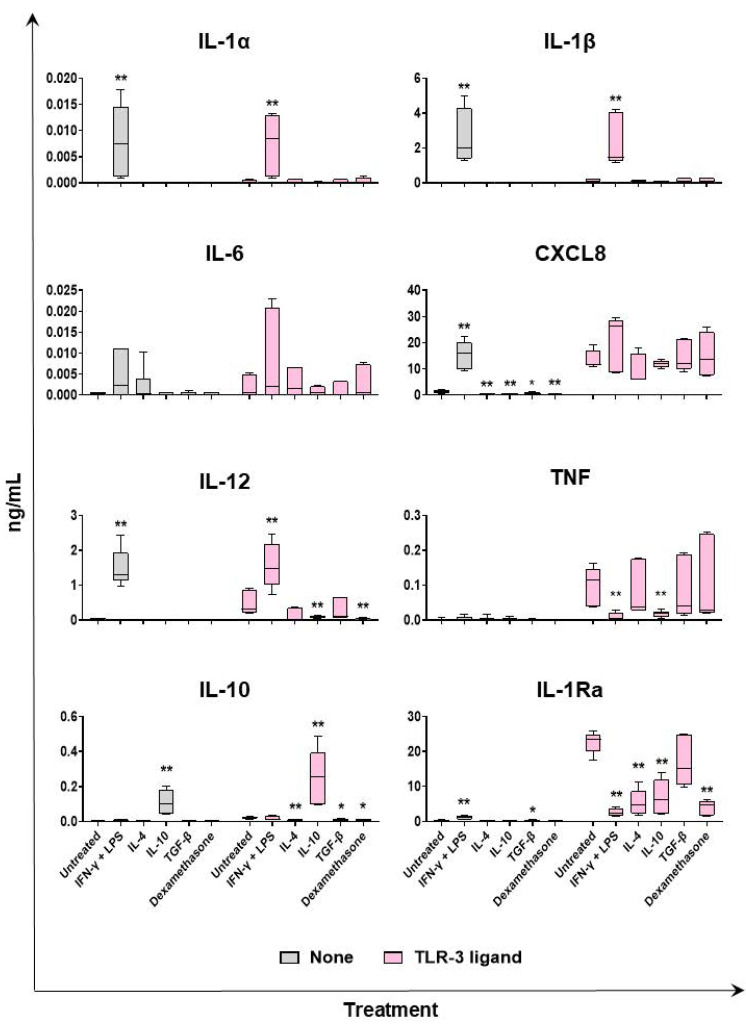
Ability of diverse macrophage subsets to release proinflammatory and anti-inflammatory cytokines in response to TLR3 agonist stimulation. The moMΦ were left untreated, or they were stimulated with diverse polarizing factors: IFN-γ + LPS (both at 100 ng/mL, moM1), IL-4 (20 ng/mL), IL-10 (20 ng/mL), TGF-β (20 ng/mL), or dexamethasone (20 ng/mL). Then, 24 h later, culture supernatants were replaced with fresh media and cells were left untreated or activated using a TLR-3 ligand (Poly I:C, 100 ng/mL); 24 h later, the amounts of IL-1α, IL-1β, IL-6, CXCL8, IL-12, TNF, IL-1Ra, and IL-10 in culture supernatants were determined using a multiplex ELISA. Data from three independent experiments utilizing different blood donors are presented. Data are displayed as box-and-whisker plots, showing the median and interquartile range (boxes) and minimum and maximum values (whiskers). For each cytokine (IL-α, IL-β, IL-6, CXCL8, IL-12, TNF, IL-1Ra, and IL-10), for both unstimulated and Poly I:C-stimulated moMΦ, values of treated macrophages were compared to the untreated control (moMΦ) using of a Mann–Whitney test. ** *p* < 0.01, and * *p* < 0.05.

## Data Availability

Data presented in the study are available on request from the corresponding author.

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
