# Peer review of "Heterogeneity of Phenotypic and Functional Changes to Porcine Monocyte-Derived Macrophages Triggered by Diverse Polarizing Factors In Vitro"

_ijms, 2023, doi:10.3390/ijms24054671_

Round 1
Reviewer 1 Report
The authors investigated the effects of classical activation (IFN-g + LPS) and M2 polarizing factors on mo-macrophage morphology, cell surface markers, cytokine expression and secretion. Furthermore, the authors also investigated the effects of TLR2 and TLR3 antagonists on the release of cytokines and chemokines from mo-macrophages. Consequently, they concluded that the plasticity of porcine macrophage is broadly comparable to that of human and mouse macrophages, highlighting several specificities of this species.
It is well written, however, there are some issues which could be improved.
The authors performed extensive experiments to clarify the effects of classical activation (IFN-g + LPS) and four M2 polarizing factors, but the results were mixed and the conclusions were not clear-cut. This is because the mo-macrophages respond differently to each of the M2 polarizing factors. On the other hand, classical activation yielded unique response patters. Therefore, at least for classical activation response, a summary of response modes should be provided in the text conclusion and/or abstract.
This study focused on cytokine, chemokine production and MHC expression. How is the phagocytotic capacity, the major role of macrophage, evaluated in this study?
In Figure 8, Figure 9, untreated mo-macrophages produced IL-6, CXCR8, and TNF in response to TLR2 antagonist and CXCR8, TNF, and IL-1Ra in response to TLR-3 antagonist. What are the characteristics of these untreated mo-macrophages cells that respond well to the TRL2 and TLR3 antagonists?
Figure 4 and Figure 5: IL-18 production by M2 polarizing factors appears to be a key finding in this study. Shouldn’t the authors also investigate IL-18 in Figure 7-9?
Figure 8 and Figure 9: There is a large variation in the values of each sample. It should be stated that there are individual differences in each animal, even if there are no differences in animal strains or sexes.
Author Response
Response to Reviewer 1 Comments
The authors investigated the effects of classical activation (IFN-g + LPS) and M2 polarizing factors on mo-macrophage morphology, cell surface markers, cytokine expression and secretion. Furthermore, the authors also investigated the effects of TLR2 and TLR3 antagonists on the release of cytokines and chemokines from mo-macrophages. Consequently, they concluded that the plasticity of porcine macrophage is broadly comparable to that of human and mouse macrophages, highlighting several specificities of this species.
It is well written, however, there are some issues which could be improved.
Point 1: The authors performed extensive experiments to clarify the effects of classical activation (IFN-g + LPS) and four M2 polarizing factors, but the results were mixed and the conclusions were not clear-cut. This is because the mo-macrophages respond differently to each of the M2 polarizing factors. On the other hand, classical activation yielded unique response patters. Therefore, at least for classical activation response, a summary of response modes should be provided in the text conclusion and/or abstract.
Response 1: As suggested by reviewer, we have provided more information regarding the generation of classically activated macrophages in vitro (Introduction Lines 50-55). In pigs, as in other species, M1 polarization can be achieved in vitro using IFN-γ and LPS, resulting in up-regulation of MHC and co-stimulatory molecules and release of pro-inflammatory cytokines (Singleton et al., 2016; Franzoni et al., 2017). Mosser et al. (2003) reported that in humans and mice, classical activation of macrophages can be achieved by exposure to two signals. The first signal is the obligatory cytokine IFN-γ, whereas the second signal is TNF (itself or an TNF inducer). TLR agonists, such as LPS, induce endogenous TNF production by the macrophage, so they are frequently used as the second signal to achieve classical activation.
Point 2: This study focused on cytokine, chemokine production and MHC expression. How is the phagocytotic capacity, the major role of macrophage, evaluated in this study?
Response 2: We agree with the author that phagocytosis is another important macrophage function. However, since two previous studies of porcine macrophages activated by classical or alternative stimuli showed no effect on phagocytic activity (Singleton et al., 2016, Sautter et al., 2018), we chose not to evaluate phagocytosis in this study.
Point 3: In Figure 8, Figure 9, untreated mo-macrophages produced IL-6, CXCR8, and TNF in response to TLR2 antagonist and CXCR8, TNF, and IL-1Ra in response to TLR-3 antagonist. What are the characteristics of these untreated mo-macrophages cells that respond well to the TRL2 and TLR3 antagonists?
Response 3: RT-qPCR data confirmed that untreated monocyte-derived macrophages expressed both TLR2 and TLR3. We have included a new Supplementary Table (S6) to show these data and the text was modified at lines 327-329, 720. The responsiveness of these cells to TLR2 and TLR3 agonists are in line with published studies (Portugal et al., 2018), including our own (Franzoni et al. 2022; Franzoni et al. 2021).
In the revised version of the manuscript, the expression of TLR2 and TLR3 in all moMФ subsets at different times post-stimulation (4, 8, 24 h) was calculated and presented in Figure S10. The relative reduced release of TNF by classically activated macrophages in response to stimulation with MagPam2Cys_P80 and Poly I:C might be linked to the reduced expression of TLR2 and TLR3 genes at the time of stimulation with TLR agonists (24 h post-treatment IFN-γ + LPS). Accordingly, the lower ability of moM(IL-4) to release TNF in response to the TLR2 agonist compared to moMФ might also be linked to the reduced expression of TLR2 at the time of treatment with MagPam2Cys_P80 lipopeptide. Text was modified accordingly at lines 343-345 and 350-352.
Point 4: Figure 4 and Figure 5: IL-18 production by M2 polarizing factors appears to be a key finding in this study. Shouldn’t the authors also investigate IL-18 in Figure 7-9?
Response 4: IL-18 production was investigated and we have included the results in the revised version of the manuscript. Release of IL-18 in response to diverse stimuli was monitored and we observed only weak release in response to stimulation with IFN-γ and LPS, but not other stimuli. Data were added to Figure 5 (Figure 4 in the revised version of the manuscript). Some discrepancy with gene expression data was noticed i.e., we observed enhanced gene expression of IL-18 in response to both IL-4 and IL-10 treatments, but no subsequent increase in IL-18 protein levels in culture supernatants. This suggest that factors at a post-transcriptional level counteract the release of this cytokine. Text was modified at lines 295-296, 315, 518-521, 538-539, and 776-777. In response to TLR2 and TLR3 stimulation, all the moMФ subsets released only small amount of IL-18, with no differences between subsets. Data were added in the supplementary files (Figure S11) and the text was modified at lines 377-378, 707.
Point 5: Figure 8 and Figure 9: There is a large variation in the values of each sample. It should be stated that there are individual differences in each animal, even if there are no differences in animal strains or sexes.
Response 5: We agree with the reviewer. Accordingly, we have added this information in the discussion section. The text was modified at lines 593-594 and 611-613.
We would like to thank the reviewer for the time spent to revised and improve our work.
Reviewer 2 Report
In the manuscript “Heterogeneity of phenotypic and functional changes to porcine monocyte-derived macrophages triggered by diverse polarizing factors in vitro”, the authors investigated porcine macrophage polarization upon exposure to different stimuli. The achieved results permit to suggest the porcine macrophages plasticity broadly comparable to human and murine macrophages and to highlight that the porcine macrophages are a good model to study inflammation.
The focus of the manuscript is new respect to literature data and very interesting.
I suggest revising the Figure 1. In fact, the authors speculate about the morphology of macrophages and report images of cytoskeleton staining. Unfortunately, in the images included in Figure 1 are very difficult to appreciate the description of the authors. So, I suggest to replace the fluorescence micrographs and to include ones at higher magnification.
Author Response
Response to Reviewer 2 Comments
In the manuscript “Heterogeneity of phenotypic and functional changes to porcine monocyte-derived macrophages triggered by diverse polarizing factors in vitro”, the authors investigated porcine macrophage polarization upon exposure to different stimuli. The achieved results permit to suggest the porcine macrophages plasticity broadly comparable to human and murine macrophages and to highlight that the porcine macrophages are a good model to study inflammation.
The focus of the manuscript is new respect to literature data and very interesting.
Point 1: I suggest revising the Figure 1. In fact, the authors speculate about the morphology of macrophages and report images of cytoskeleton staining. Unfortunately, in the images included in Figure 1 are very difficult to appreciate the description of the authors. So, I suggest to replace the fluorescence micrographs and to include ones at higher magnification.
Response 1: In Figure 1b, fluorescent microscopy images were acquired using a 40x objective magnification, which was the highest possible magnification. In the revised version of the manuscript, fluorescence microscopy images were moved to supplementary materials, and we modified the results and discussion section, focusing on physical characteristics (FSC and SSC parameters) evaluated with flow cytometry. Text was modified at lines 92-100, 108-112, and 436-442.
We would like to thank the reviewer for the time spent reviewing our work and for his/her positive feedback.
Reviewer 3 Report
The manuscript represents a high quantity of unorganized and random data regarding macrophage polarization in pigs. The first issue corresponds to figure 1. It is very difficult to believe that the agonists used do not change the morphology of monocytes in culture. This figure does not provide important information and should be part of the supplementary figures, and please do not repeat the same figures of the manuscript with supplementary information. The second issue is why almost all the peripheral monocytes express CD16 if it is a classic marker of inflammatory macrophages; IL10 does not induce CD16. Why do the authors use those concentrations of stimulants? Is there a dose-response curve?
CD14, the antibody used against human CD14, is probably unsuitable for detection. The other issue is why CD86 and other Fc receptors were not analyzed. Figure 3 should be in the text and figure 2 should be omitted, Figure 4 is confusing with IFN gamma and LPs is difficult to believe there is no induction of TNF and mild enhancement of IL-6. Please check the data carefully. The differences shown in figures 5 and 6 are also difficult to believe.
After the rearrangement of the manuscript, the discussion should be rewritten, as several conclusions about subpopulations of monocytes are not supported by the data.
In general, the article must be rewritten based on the reanalysis of the data
Author Response
Response to Reviewer 3 Comments
Point 1: The manuscript represents a high quantity of unorganized and random data regarding macrophage polarization in pigs. The first issue corresponds to figure 1. It is very difficult to believe that the agonists used do not change the morphology of monocytes in culture. This figure does not provide important information and should be part of the supplementary figures, and please do not repeat the same figures of the manuscript with supplementary information.
Response 1: In the revised version of the manuscript, fluorescence microscopy images were moved to supplementary materials, as also suggested also by Reviewer 2. We subsequently reduced the results and discussion section, focusing more on physical characteristics measured by flow cytometry. The macrophage subsets presented no statistically significant differences in terms of forward and side scatter characteristics, which were in agreement with previous publications (Carta et al., 2021; Garcia-Nicolas et al., 2014; Franzoni et al., 2017). Text was modified at lines 92-100, 108-112, and 436-442.
Point 2: The second issue is why almost all the peripheral monocytes express CD16 if it is a classic marker of inflammatory macrophages; IL10 does not induce CD16.
Response 2: The vast majority of porcine moMФ express CD16, in agreement with previous work (Stepanova et al., 2022, Carta et al., 2021, Franzoni et al., 2022). We observed that CD16 expression was further upregulated by IL-10 treatment, as observed in our previous publication (Carta et al., 2021) and in agreement with human macrophage studies (Pahl et al., 2014, Wang et al., 2011). We have included a new reference to upregulation of CD16 expression on human macrophages following IL-10 stimulation to the discussion section [Wang et al., 2011; 28] (Line 464).
Point 3: Why do the authors use those concentrations of stimulants? Is there a dose-response curve?
Response 3: We chose these concentrations based on previously published studies in pigs. In our previous work, we used 100 ng/mL of IFN-γ and 100 ng/mL of LPS to generate moM1, whereas we used 20 ng/mL of IL-4 to generate moM2 (Franzoni et al., 2017; Franzoni et al, 2017; Franzoni et al., 2020). These concentrations were similar to those used by other groups working on porcine macrophages (Garcia-Nicolas et al., 2014; Singleton et al., 2016). We added some of these references in the materials and methods section of the revised version of the manuscript (Line 662). In other studies performed to better characterize alternative activation of porcine macrophages, we used for IL-10 and TGF-β the same concentrations as used for IL-4 i.e., 20 ng/mL (Carta et al., 2021; Franzoni et al., 2022). More references were also added in the material and methods section at line 664. In this work, all M2-related polarizing factor (IL-4, IL-10, TGF-β, dexamethasone) were used at the same concentration (20 ng/mL), to exclude the possibility that differences observed between M2-like subsets were linked to the dosage used.
Point 4: CD14, the antibody used against human CD14, is probably unsuitable for detection.
Response 4: The Tuk4 monoclonal antibody recognizes a well conserved epitope on CD14 showing cross-reactivity with a range of mammalian species including pigs (Jacobsen et al., 1993). We have used the Tuk4 mAb in the past to study porcine dendritic cells and macrophages (Franzoni et al., 2021; Carta et al., 2021; Edwards et al., 2017; Singleton et al., 2018; Franzoni et al., 2017; Soldevila et al., 2018; Soldevila et al., 2021). Tuk4 has also been used by other groups to label porcine CD14 e.g., Chamorro et al., 2005; Byrne et al., 2020; Pujol et al., 2021. We added this information in supplementary materials and Jacobsen et al., 1993 among the references [66]. Text was slightly modified also in the materials and methods section (line 689).
Point 5: The other issue is why CD86 and other Fc receptors were not analyzed. Figure 3 should be in the text and figure 2 should be omitted,
Response 5: We agree with the suggestion to remove Figure 2, and we have moved this to the supplementary materials. We chose to exclude CD86 from the analysis since a previous study had already shown that expression of the CD80/CD86 complex was upregulated on porcine monocyte-derived macrophages by IFN-γ/LPS but not by M2 polarizing factors (IL4, IL10, or dexamethasone) (Singleton et al., 2016). We added this information in the text (lines 426-429). We thank the reviewer for the suggestion to also consider other Fc receptors, which we will consider for future studies.
Point 6: Figure 4 is confusing with IFN gamma and LPs is difficult to believe there is no induction of TNF and mild enhancement of IL-6. Please check the data carefully. The differences shown in figures 5 and 6 are also difficult to believe.
Response 6: Data in Figure 4 (Figure 3 in the revised version of the manuscript) are also reported in supplementary materials (Table S4). MoM1 presented higher expression of both TNF and IL-6 compared to moMФ, with a fold-change of 12.06 and 121.27, respectively. There experiments were performed using three diverse blood donor pigs and the P values for these data were 0.096566 and 0.099971, respectively. In the revised version of the manuscript, we performed RT-qPCR to evaluate TNF and IL6 expression in moMФ from five diverse blood donor pigs and we observed that moM1 presented higher expression of both cytokines, especially IL-6. These data were added in supplementary materials (Figure S9) and text was modified at lines 242-244. Table S5 was also modified. The data presented in Figures 5 and 6 were made from technical duplicate samples from moMФ isolated from five donor pigs. We were only allowed to use a relatively small number of blood donor pigs for the whole project under authorization n° 1232/2020-PR.
Point 7: After the rearrangement of the manuscript, the discussion should be rewritten, as several conclusions about subpopulations of monocytes are not supported by the data.
In general, the article must be rewritten based on the reanalysis of the data
Response 7: In the revised version of the manuscript, the results and discussion were modified in response to the useful comments raised by reviewers. The text of the result section was modified at line 92-100, 242-244, 295-296, 327-329, 343-345, 350-352, 377-378, 398-399. The discussion and conclusion section was modified at line 426-429, 436-442, 464, 518-521, 538-539, 593-594, 611-613, 776-777.
We would like to thank the reviewer for the time spent reviewing our work and we hope to have improved it in this revision stage.
Round 2
Reviewer 3 Report
The authors modified the manuscript as requested. I can not be suitable for publication